

# Limits on the ability of global Eulerian models to resolve intercontinental transport of chemical plumes

Sebastian D. Eastham[1], Daniel J. Jacob[1]

[1]School of Engineering and Applied Sciences, Harvard University, Cambridge, MA 02138, USA

*Correspondence to*: Sebastian D. Eastham (seastham@fas.harvard.edu)

**Abstract.** Quasi-horizontal chemical plumes in the free troposphere can preserve their concentrated structure for over a week, enabling transport on intercontinental scales with important environmental impacts. Global Eulerian chemical transport models (CTMs) fail to preserve these plumes due to fast numerical dissipation. This work examines the causes of this dissipation and how it can be cured. GEOS-5 meteorological data at 0.25°×0.3125° horizontal resolution and ~0.6 km vertical resolution in

the free troposphere are used to drive a worldwide ensemble of GEOS-Chem CTM plumes at resolutions from 0.25°×0.3125° to 4°×5°, in both 2-D (horizontal) and 3-D. 2-D simulations enable examination of the sensitivity of numerical dissipation to grid resolution. We show that plume decay is driven by flow divergence and shear, filamenting the plume until GEOS-Chem's high-order advection scheme cannot resolve gradients and fast numerical diffusion ensues. This divergence, measured by the Lyapunov exponent ($\lambda$), is much stronger at extratropical latitudes than in the tropics, resulting in accelerated decay. The plume

decay constant ($\alpha$) is linearly related to $\lambda$, and increasing grid resolution provides only modest benefits toward plume preservation. 3-D simulations show near-complete dissipation of plumes within a few days, independent of horizontal grid resolution and even in the tropics. This is because vertical grid resolution is inadequate in all cases to properly resolve plume gradients. Increases in vertical grid resolution in the free troposphere should be prioritized over horizontal resolution if intercontinental transport in global Eulerian models is to be resolved.

## 1 Introduction

Global transport of pollution mainly takes place in the free troposphere where winds are strong and pollutant lifetimes are long. Much of this transport takes place in well-defined, concentrated layers or plumes that can remain coherent for a week or more and affect intercontinental scales. Models fail to reproduce these persistent plumes due to rapid dissipation by numerical diffusion. Here we use the GEOS-Chem chemical transport model (CTM) to understand this problem.


The free troposphere, defined as the region between the turbulent planetary boundary layer (PBL) and the quiescent stratosphere, experiences strong wind shear (divergence) in a convectively stable environment. Stability allows the formation of persistent laminae (layers) or plumes, first detected by early radiosonde measurements (Danielsen 1959) and later shown to be ubiquitous throughout the free troposphere (Newell et al. 1999; Thouret et al. 2000). The plumes are quasi-horizontal,

typically fanning out over hundreds of kilometers with a vertical thickness on the order of 1 km (Colette et al. 2005; Heald et





al. 2003; Hudman et al. 2004; Jaffe et al. 2003; Liang et al. 2007; Mauzerall et al. 1998; Stoller et al. 1999). Free tropospheric plumes resulting from stratospheric intrusions can retain 150 ppb of ozone over a period of weeks (Trickl et al. 2011). Such global-scale transport with little dilution has important implications for environmental impacts, interactions with weather, and chemical aging.

Eulerian models used for simulating global atmospheric transport fail to reproduce this persistent layered structure. The modeled plumes dissipate within days by mixing with the background (Heald et al. 2003; Vuolo et al. 2009). Eulerian models simulate transport as a flux divergence for fixed grid cells, and the rapid dissipation implies a large transportation error from numerical diffusion even when a highly accurate advection algorithm is used (Rastigejev et al. 2010). A Lagrangian approach,

where transport is calculated for individual air parcels carried by with the flow with no interaction between neighbors, would avoid this problem (Khosrawi et al. 2005). Global Lagrangian models have been used with success in the stratosphere to describe the sharp gradients at the edge of the polar vortex (Fairlie et al. 1999; Hoppe et al. 2014; Konopka et al. 2003). However, they are not used in global tropospheric applications because of difficulties in dealing with convective motions, homogeneity of coverage, and non-linear chemistry (Brasseur and Jacob 2016). Adaptive mesh refinement techniques have

shown promise in addressing this issue in Eulerian models (Semakin and Rastigejev, 2016) but are computationally complex. There is a need to understand why persistent free tropospheric plumes are so rapidly dissipated in Eulerian models, and how this behavior can be corrected.

A theoretical study by Rastigejev et al. (2010) examined the causes of the fast numerical dissipation of intercontinental plumes

in Eulerian models. Numerical diffusion is due to finite differencing of the advection equation on the model grid such that the gradients between grid cells are imperfectly described. The order of a numerical advection scheme is defined by the number of adjacent grid cells used to resolve a local gradient. Rastigejev et al. (2010) showed that a highly accurate, third-order, finite-volume advection scheme such as is used in GEOS-Chem (Lin and Rood 1996) successfully preserves plume structures for over 10 days in a uniform flow, but fails rapidly when real-world divergence is applied. Flow divergence acts to filament,

stretch, and thin the plume until it is resolved by only a few grid cells. At that point the gradient can no longer be represented with a high-order scheme; the scheme collapses to first order, resulting in very fast numerical dissipation. Increasing grid resolution only delays the onset of this effect, and may amplify it through the additional flow divergence. Rastigejev et al. (2010) proposed that the plume dissipation rate in a stretched flow is eventually set by the Lyapunov exponent $\lambda = \partial u/\partial x$ of the flow, defined as the exponential rate at which adjacent trajectories (aligned with the wind speed vector $\mathbf{u}$) diverge from

each other. Increasing the grid resolution $\Delta x$ of the model would then slow down the rate of decay only as $\Delta x^{0.5}$, rather than $\Delta x^3$ as might be expected from a third-order advection scheme. Given that computational costs in Eulerian models typically rise at a rate of $\Delta x^2$ to $\Delta x^3$, this implies that a computationally expensive increase in resolution will yield only marginal reductions in the errors driving numerical plume dissipation.





In this paper, we examine whether the theory of Rastigejev et al. (2010) can explain the fast numerical decay of free tropospheric plumes in Eulerian models, and what the implications are for curing this problem through increasing grid resolution. We use for this purpose global 2-D (horizontal) and 3-D versions of GEOS-Chem to simulate atmospheric flow at horizontal grid resolutions ranging from $0.25° \times 0.3125°$ ($\sim 25 \times 30$ km$^2$) to $4° \times 5°$ ($\sim 400 \times 500$ km$^2$) and with a native vertical resolution of $\sim 0.6$ km in the free troposphere. We quantify the decay rate for plumes originating in different locations around the world, relate this decay rate to flow stretching, and conclude as to the potential to preserve the plumes through the use of improved grids.

## 2 Theory

The theory presented by Rastigejev et al. (2010) for numerical diffusion of stretched plumes begins with the advection equation (1)

$$\frac{\partial n}{\partial t} + \nabla \cdot (n\boldsymbol{u}) = 0 \tag{1}$$

where $n$ is the number density of an inert chemical ("tracer") and $\boldsymbol{u}$ is the wind vector. Rastigejev et al. (2010) expressed this equation in its advective form (2) and included a diffusivity term $D$ to describe numerical diffusion:

$$\frac{\partial C}{\partial t} + \boldsymbol{u} \cdot \nabla C = D\nabla^2 C \tag{2}$$

Here $C = n/n_a$ is the volume mixing ratio (VMR) and $n_a$ is the number density of air. This form explicitly accounts for the effect that numerical diffusion has on the modeled flow. Without this numerical diffusion, the advection equation would strictly conserve the VMR. When numerical diffusion is included, the VMR decays over time as the plume dissipates.

Let us consider now the conceptual picture of a model plume with uniform VMR diluting by numerical diffusion into a background atmosphere with a VMR of zero. The plume has surface area $S$ and volume $V$. Mass balance for the plume is given by

$$V\frac{dC}{dt} = -DS\boldsymbol{k} \cdot \nabla C \tag{3}$$



where $\boldsymbol{k}$ is a unit vector normal to the plume surface. Rastigejev et al. (2010) defined a characteristic length $r_b$ for decay across the edge of the plume so that $\nabla C \sim C/r_b$. They further defined a characteristic width of the plume as $W = V/S$. Thus equation (3) becomes

$$\frac{dC}{dt} = -\left(\frac{D}{r_b W}\right) C \tag{4}$$

This implies an exponential decay in $C$, such that

$$C(t + \Delta t) = C(t)\exp(-\alpha \Delta t) \tag{5}$$

where the decay constant $a$ is given by

$$\alpha = \frac{D}{r_b W} \tag{6}$$

and $\Delta t$ is some time interval.

The decay rate of the plume is proportional to the numerical diffusivity $D$, which is dictated by the order $f$ of accuracy of the numerical advection scheme: $D \sim \Delta x^f$. However, the decay rate also depends on $r_b$. In a divergent flow, stretching of an initially broad plume ($W \gg r_b$) causes $r_b$ to decrease, and hence the decay rate to increase. This stretching can be represented by the Lyapunov exponent, defined as

$$\lambda = -\frac{\partial v}{\partial y} = \frac{\partial u}{\partial x} \tag{7}$$

20   where $(u,v)$ are the $(x,y)$-components of the velocity and $x$ is taken as the direction of stretching. The Lyapunov exponent defines the exponential rate constant at which initially adjacent trajectories diverge.

Stretching in a divergent flow thins the plume, while numerical diffusion thickens it. Under constant divergence (constant $\lambda$), an equilibrium size for $r_b$ is reached when these two processes proceed at the same rate. Since the rate constant for diffusion is $\sim D/r_b{}^2$, and the rate constant for stretching is $\lambda$, equilibrium is reached when



$$\frac{D}{r_b^2} = \lambda \tag{8}$$

Replacing into (6), we find

$$\alpha = \frac{\sqrt{D\lambda}}{W} \tag{9}$$

Thus the rate of decay in a stretched flow is less sensitive to $D$ than expected. Furthermore, if the plume has stretched to be only a few gridboxes thick, then the gradient across the plume boundary cannot be resolved by a high-order scheme anymore and any numerical advection scheme collapses to first order (Godunov 1959). Under these conditions $D \sim \Delta x$ and thus $a \sim \Delta x^{0.5}$; the decay rate improves only as the square root of the grid resolution.

We also see from equation (9) that the decay rate increases with the rate of stretching as measured by $\lambda$. Regions of divergent flow are expected to experience faster plume decay. Eventually, for a fully stretched plume we have $W = r_b$. Under these conditions, equations (8) and (9) yield $a = \lambda$ and the decay rate is independent of the grid resolution - a remarkable result.

The Rastigejev et al. (2010) theory thus paints the following picture for the model decay of a free tropospheric plume in a divergent flow (as is realistically found in the atmosphere) and its dependence on grid resolution $\Delta x$. A plume that is initially well-resolved on the model grid will decay with a rate constant propotional to $\Delta x^f$, where $f$ is the order of accuracy of the numerical advection scheme. As the plume stretches, shears and filaments, it becomes poorly resolved on the model grid, and at that point the decay rate becomes proportional to $(\lambda\Delta x)^{0.5}$, i.e. only weakly responsive to increasing grid resolution and dependent on the stretching rate. In fact, increasing grid resolution may increase the stretching of the flow by introducing additional convergence-divergence zones that would be averaged out at coarser resolution. Under those conditions Rastigejev et al. (2010) find that the decay rate may be proportional to $\Delta x^{0.25}$, an even weaker grid resolution dependence. Eventually, the filamented plume decays with a rate constant defined by $\lambda$ and at that point very fast dissipation takes place that is resolution-independent. This theory, if correct, has major implications for understanding the decay of free tropospheric plumes in Eulerian models, and the value of increasing grid resolution. In what follows we test the theory using global simulations in actual atmospheric flow with the GEOS-Chem CTM.



## 3 Testing theory with the GEOS-Chem CTM

We simulate transport of free tropospheric plumes in v11-01e of the global Eulerian GEOS-Chem CTM originally described by Bey et al. (2001). The model is driven by winds and other meteorological data archived every 3 hours from the Data Assimilation System of the NASA Goddard Earth Observing System (GEOS-5) with $0.25° \times 0.3125°$ horizontal resolution on 72 vertical levels. The vertical grid resolution in the free troposphere between 4 and 8 km altitude is about 0.6 km. We apply the model to an inert chemical tracer with only advection enabled. Subgrid transport processes, including convection and boundary-layer mixing, are disabled. Thus the model only solves for advection (equation (1)), using the 3-hour GEOS-5 FP archive of mean horizontal winds and instantaneous surface pressure. Horizontal winds are adjusted with a "pressure fixer" (Horowitz 2003) to ensure consistency with the 3-hour pressure change. Vertical winds are derived from divergence of the horizontal winds and the change in surface pressure.

Horizontal advection is calculated using the FFSL-3 finite volume scheme developed by Lin and Rood (Lin and Rood 1996) and commonly called "tpcore". This scheme uses the monotonic piecewise- parabolic method (PPM) when the Courant- Friedrichs- Lewy number (CFL) is less than or equal to one, and a semi-Lagrangian method for CFL > 1. A semi-monotonic PPM is used in the vertical direction with the enforcement of Hyunh's second monotonicity constraint. The FFSL- 3 scheme is formally third-order accurate in space, such that increasing the grid resolution $\Delta x$ by a factor of 2 should reduce numerical errors by a factor of 8.

We conduct 2-D (horizontal) and 3-D simulations at 5 different horizontal resolutions: $0.25° \times 0.3125°$ (native), $0.5° \times 0.625°$, $1° \times 1.25°$, $2° \times 2.5°$, and $4° \times 5°$. 2-D simulations allow an analysis of the effect of horizontal resolution over a factor of 16 (from $0.25° \times 0.3125°$ to $4° \times 5°$) to test the theoretical dependences of plume dissipation on grid resolution derived by Rastigejev et al. (2010). We cannot carry out a similar analysis in the vertical because the ~0.6 km native vertical resolution of the GEOS-5 data in the free troposphere is too coarse. In all cases, winds and pressures are retrieved from the GEOS-5 FP archive at the $0.25° \times 0.3125°$ native resolution and subsequently averaged spatially to drive coarser-resolution simulations as is routinely done in GEOS-Chem (Bey et al. 2001; Philip et al. 2015). A dynamical timestep of 5, 5, 10, 15, and 30 minutes respectively is used for each resolution. Decreasing the timestep to 5 minutes for all simulations had no significant effect. All concentrations shown and discussed are based on instantaneous VMRs stored at one-hour intervals.

2-D simulations are performed by taking the pressure-weighted average of the wind velocity in each atmospheric column and setting the surface pressure tendency to zero. Although clearly idealized, there is some realism to the 2-D simulations in that free tropospheric layers are vertically stratified and most of the shearing and dissipation can be expected to take place in the horizontal. Most relevantly, the 2-D simulations allow us to test the theory of Rastigejev et al. (2010) for the sensitivity of plume dissipation to grid resolution.



We conduct simulations of the first 9 days of July 2015 for plumes initialized in different locations around the world with a homogeneous unit mixing ratio over a cuboid 12° in latitude by 15° in longitude, and zero outside. This size is chosen so that the initial plume is coarsely resolved at 4°×5° but finely resolved at 0.25°×0.3125°. 90 non-interacting plumes are initialized over the global domain (Fig. 1) to examine how location affects numerical diffusion. In our 2-D simulations, only horizontal advection is enabled, and the model domain is restricted to a single vertical layer. In our 3-D simulations, vertical advection is enabled, and each plume is initialized at 3.9 km pressure altitude over a single GEOS-5 model level that is 470 m thick.

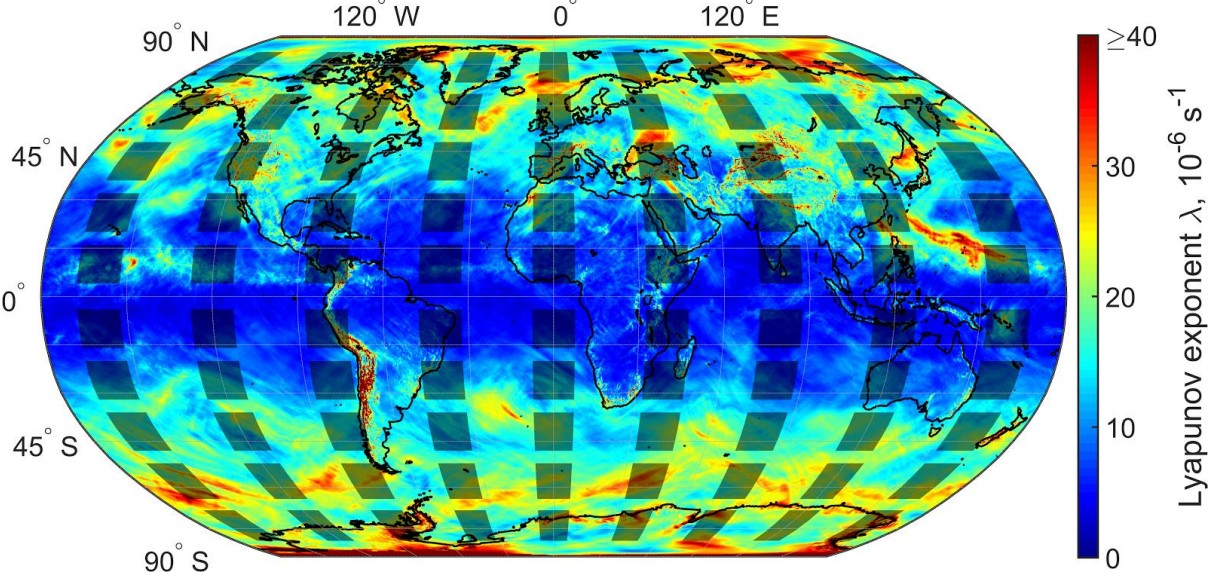

**Figure 1. Plume initialization locations overlaid on the mean 2-D Lyapunov exponent _λ_ for the July 1-9, 2015 period of the simulations. Initialization regions for each of the 90 plumes are shown as semi-transparent black boxes.**

## 4 Quantifying numerical diffusion and stretching

### 4.1 Numerical diffusion

Exact solution of the advection equation translates mixing ratios downwind without altering them. In other words, initial mixing ratios in a plume remain constant as the plume is advected downwind. Any plume decay in our advection-only simulation must be the result of numerical diffusion. In the real atmosphere, plumes decay by molecular diffusion that operates on millimeter scales and is the end result of the turbulent eddy cascade that filaments the plume into finer and finer strands (Brasseur and Jacob 2016). This subgrid turbulence is particularly fast for vertical mixing in the boundary layer, and is typically parameterized in models with a turbulent diffusion scheme. It is usually ignored in the horizontal direction or in the free troposphere, under the assumption that spurious numerical diffusion effectively carries out the mixing.





Numerical diffusion arises from finite differencing over grid cells when solving the advection equation. Odd-order schemes such as the PPM tend to introduce diffusion, artificially smoothing the solution in areas with sharp concentration gradients. Even-order schemes tend instead to be dispersive, producing spurious oscillations, and this artifact information is even less desirable than numerical diffusion. Higher-order schemes also tend to produce spurious oscillations when there are

discontinuities in the concentration field (Godunov 1959; Brasseur and Jacob 2016). To get around this, modern advection schemes such as FFSL-3 employ flux limiters that locally reduce the scheme to first order in the vicinity of discontinuities. This prevents spurious oscillations at the cost of increasing numerical diffusion.

Numerical diffusion in GEOS-Chem is illustrated in Fig. 2 with an example of a 2-D plume at $1° \times 1.25°$ grid resolution. The

plume decays with time due to numerical diffusion. This decay is reflected by an increase in the plume area and a decrease in the plume VMR. The rate of decay can be measured by the reduction in the plume maximum VMR, as done by Rastigejev et al. (2010) and expressed in equation (5) in terms of an exponential plume decay constant $a$. In situations where the plume is resolved by a large number of grid cells, the maximum VMR can be buffered from the effects of numerical diffusion by surrounding grid cells, even as the plume frays at its edges. In the example shown here, the maximum value remains nearly

unchanged for 4 days as a result of this buffering and hence $a$ is near zero. Even beyond 4 days, $a$ can be highly variable depending on the local flow divergence (Rastigejev et al., 2010) and tends to decrease as the plume dissipates because dissipation smooths the VMR gradient.

An alternate metric of numerical diffusion is the size of the plume. The thick red contour in Fig. 2 shows the minimum area

containing 90% of the total mass of tracer in the plume. As the simulation progresses, diffusion of the plume increases this area. We define the square root of this area as the characteristic size of the plume, normalized by the value at plume initialization. In 3-D, the plume size is taken as the cube root of the total volume occupied by 90% of the tracer mass, after accounting for differences in air density. Plume size is a more sensitive indicator of plume diffusion, as shown for our example in the bottom panel of Fig. 2, because it accounts for the fraying at the plume edges and is a smoother function than $a$. Here

we will mostly use the decay constant $a$ of maximum VMR as metric of plume decay, for consistency with Rastigejev and to compare to theory (Section 2), but we will also show some results for plume size.





**Figure 2. Numerical diffusion of a 2-D (horizontal), inert plume in GEOS-Chem for a 9-day simulation at 1°×1.25° horizontal resolution. This particular plume was initialized between Australia and New Zealand (Fig. 1). The red contour shows the minimum area containing 90% of the tracer mass. From top to bottom, the lower three panels show: the normalized maximum VMR in the plume; the 6-hour moving average decay constant α calculated from equation (5); and the plume size, defined as the square root of the 90% contour area and normalized by the initial value.**

### 4.2 Stretching

Plume stretching can be quantified by the local Lyapunov exponent of the flow, as defined in equation (6) for horizontal stretching. Rastigejev et al. (2010) calculated this Lyapunov exponent using a level-set approach (Leung 2011). Here we calculate an approximately equivalent quantity. If $\delta(t)$ is the separation of two adjacent points at time $t$, then after a time interval $\Delta t$





$$|\delta(t + \Delta t)| \simeq \exp(\lambda)|\delta(t)| \tag{11}$$

Rearranging equation 11 gives

$$\lambda = \frac{1}{\Delta t} \ln\left(\frac{|\delta(t + \Delta t)|}{|\delta(t)|}\right) \tag{12}$$

This can be directly applied in an Eulerian model framework, acknowledging the separate treatment of winds in each dimension. For taxicab geometry, as in the orthogonal latitude-longitude discretization, we measure the separation $\delta(t)$ as ($\Delta x + \Delta y$) at time $t = 0$, where $\Delta$ denotes the grid cell spacing. The separation at time $t + \Delta t$ is then given by

$$|\delta(t + \Delta t)| = (\Delta x + |\Delta t \Delta u|) + (\Delta y + |\Delta t \Delta v|) \tag{13}$$

Here, $\Delta u$ and $\Delta v$ refer to the change in wind speed between the current grid cell and the downstream cell in each direction. Using the absolute value prevents flow stretching in one direction from being offset by compression in another. Instead, this metric responds to flow stretching in either the $x$- or $y$-direction. We also assume that the change in separation will be small relative to the initial grid spacing, allowing us to approximate $\ln(x) \approx (1-x)$. Replacing into equation (12) and denoting the change of wind speed in the direction of motion as $\Delta u$ or $\Delta v$, we find

$$\lambda \simeq \frac{|\Delta u| + |\Delta v|}{\Delta x + \Delta y} \tag{14}$$

At each grid cell, the above equation is applied to yield an estimator for the rate of horizontal flow stretching. Figure 1 displays the mean Lyapunov exponents at 0.25°×0.3125° for the full 9-day simulation period in the 2-D flow. As discussed by Stohl (2001), the weaker synoptic-scale eddies at low latitudes result in less flow stretching relative to the higher latitudes.

**5 2-D plume decay and relation to stretching**

**5.1 Sensitivity to grid resolution**

Figure 3 shows the evolution of the plume peak VMR and decay constant in the 2-D simulations as a function of latitude for different grid resolutions.



The rate of plume decay increases with latitude. A tropical plume with initial area of 12°×15° retains over 99% of its original maximum VMR after 9 days at a resolution of 0.25°×0.3125°, 98% at 0.5°×0.625°, and 89% at 1°×1.25°. Outside of the tropics, these values fall to 82% at 0.25°×0.3125°, 59% at 0.5°×0.625° and 38% at 1°×1.25°. A plume in the tropics is better preserved at 1°×1.25° than a plume outside the tropics at 0.25°×0.3125°.

Rastigejev et al. (2010) presented a single example of a Chinese plume transported over the Pacific in 2-D flow as illustration of their theory. Starting from the same initial 12°×15° plume dimension, they found the maximum VMR to drop to 10% of its original value after 9 days 1°×1.25°. Our results considering a large ensemble of plumes do not show such a drastic decay. In fact, it would seem that the 0.25°×0.3125° resolution is largely successful at preserving plumes over the 9-day period. As we will see in section 6, this success does not hold for 3-D plumes; but we focus on 2-D plumes in this section to better understand dependences on flow stretching and grid resolution.

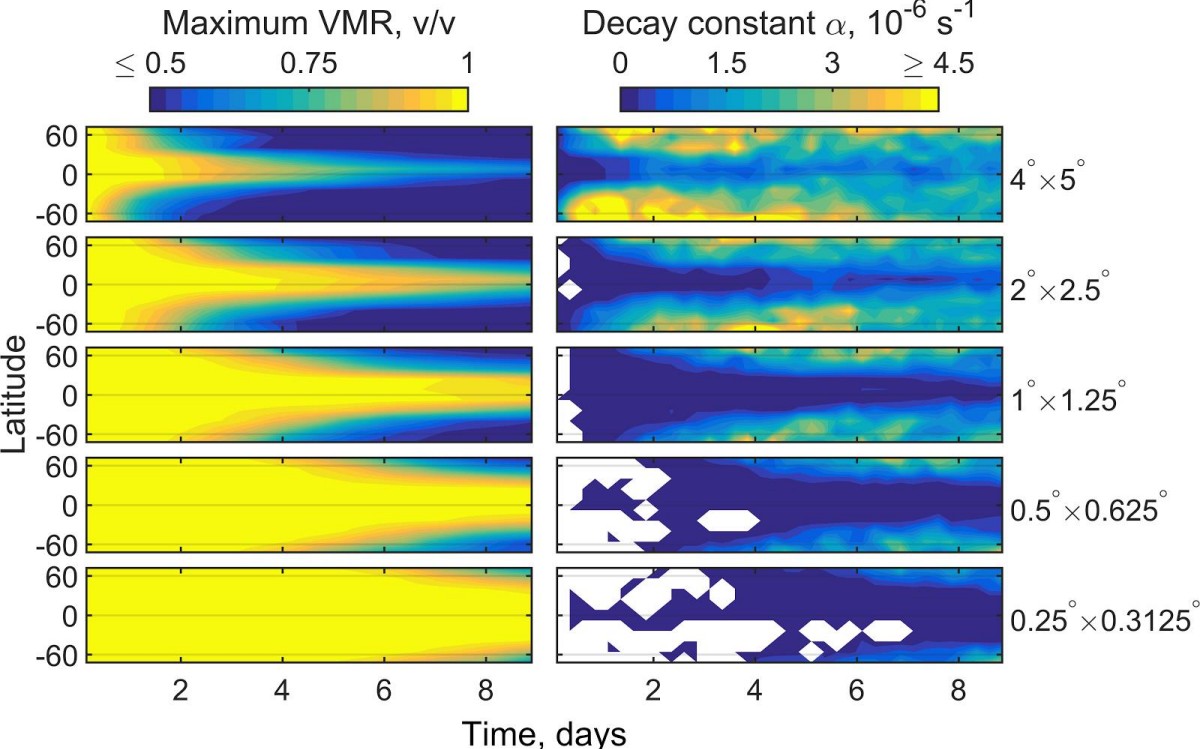

**Figure 3. Evolution with time of the maximum VMR and the decay constant $a$ for GEOS-Chem plumes initialized at different latitudes in a 2-D (horizontal) flow. White areas in the right panels correspond to no significant plume decay ($a \approx 0$).**

## 5.2 Relationship to flow divergence

Following the theory of Rastigejev et al. (2010) as summarized in Section 2, we examined the relationship between plume decay (as represented by the plume decay constant $a$) and flow stretching (as represented by the Lyapunov exponent $\lambda$). Figure





shows the relationship between $a$ and $\lambda$ for our ensemble of plumes at 0.25°×0.3125°, 1°×1.25°, and 4°×5° grid resolutions. Each datapoint corresponds to the decay constant for one of the 90 plumes (Fig. 1) at a given resolution, averaged over the first 200 hours of the simulation. Values of $\lambda$ tend to be smaller at coarser resolutions because small-scale convergent-divergent circulations average out.

Results in Fig. 4 show in general a strong correlation between $a$ and $\lambda$, demonstrating that flow stretching plays a major role in driving plume decay. The relationship is linear at 4°×5° grid resolution, as would be expected for a fully stretched plume (section 2). At the higher resolutions, $a$ remains near zero when $\lambda$ is low (tropical plumes) implying that the plume remains well-defined when stretching is weak. However, when stretching is strong ($\lambda > 10^{-5}$ s$^{-1}$), we still find a linear relationship

between $a$ and $\lambda$, indicating fast plume decay from plume filamentation even at the higher resolutions. Rastigejev et al. (2010) gave $a = \lambda$ for the fully stretched plume on the model grid, but we find weaker slopes that decrease with increasing resolution (Fig. 4). This suggests that the model actually retains some capability to describe cross-plume gradients, and the Rastigejev et al. (2010) assumption of a sharp discontinuity on the model grid must be viewed as a limiting case.

The difference in the slope of the regression lines gives an approximate measure of the improvement gained by increasing resolution by a factor of 4. Increasing resolution provides a "delay" in terms of the minimum $\lambda$ at which the plumes being to decay rapidly due to stretched-flow diffusion. At high resolution and low $\lambda$, the maximum VMR is preserved for the full 200 hours, and minimal numerical diffusion occurs. At high $\lambda$ or low resolution, the buffering is insufficient, and numerical diffusion proceeds at a rate that decreases by about a factor of 2 for every 4-fold increase in resolution This supports a central

result of Rastigejev's theory that the plume decay rate decreases as the square root of the grid resolution.

The right-hand panel of Fig. 4 shows the response of the plume size to diffusion after 200 hours. At 4°×5°, the average size increase after 200 hours is a factor of 4.5, compared to 2.0 at 0.25°×0.3125°. Unlike the decay constant α which relates to the maximum VMR, the plume size consistently increases as λ increases, reflecting the effect of stretching in fraying the plume

edges even if the plume maximum VMR is preserved. The rate of improvement in plume size with resolution is therefore slower and there is no minimum value of $\lambda$, as numerical diffusion affects the plumes immediately at all resolutions. Overall, a factor of 16 increase in grid resolution yields a reduction in plume size by a factor of about 3, compared to a factor of 4 for the decay constant.



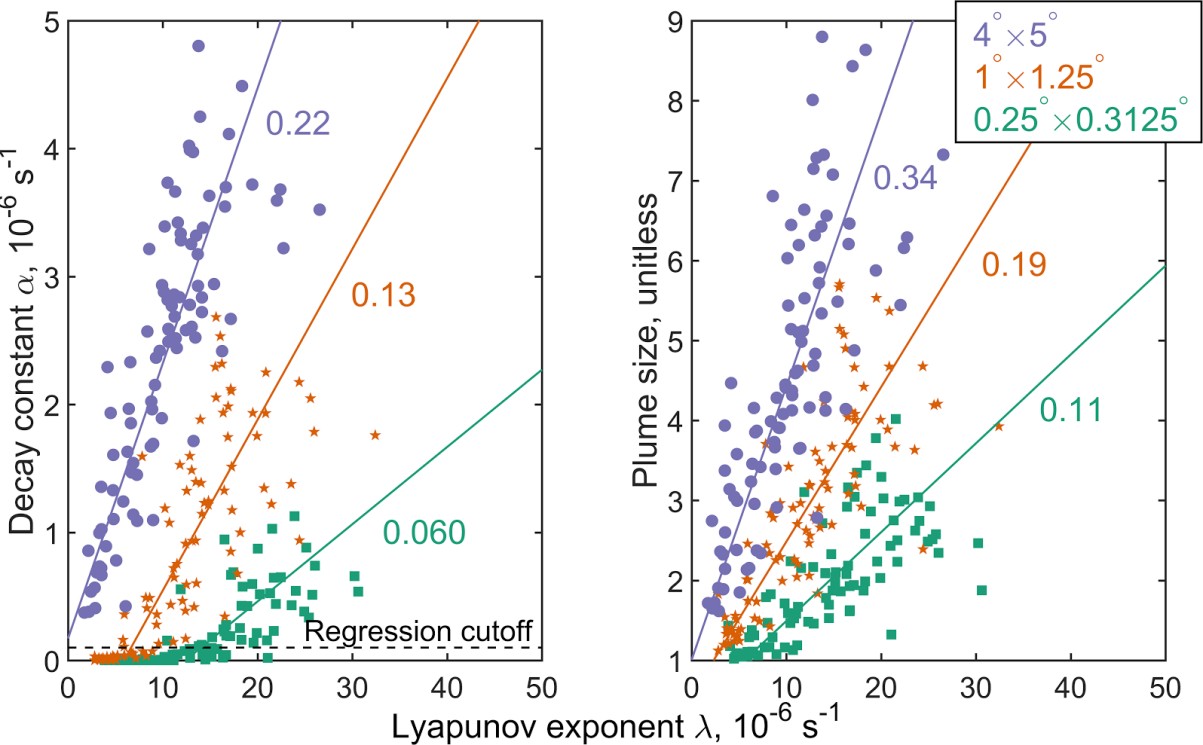

**Figure 4. Dependence of plume decay on the stretching of the atmospheric flow as measured by the Lyapunov exponent. The figure shows the average plume decay constant *a* over 9 days of aging, and the plume size after 9 days, in a 90-member ensemble of 2-D plume simulations worldwide (Fig. 1). Each datapoint corresponds to a single plume and the grid resolution is identified by color. Linear regression lines are calculated using reduced major axis regression, discounting points below the "regression cutoff" line. The slopes are shown next to the regression lines. Values of $r^2$ are in the range 0.49 to 0.62 for all regressions.**

**5.3 Effective order of accuracy**

The PPM advection scheme used in GEOS-Chem is 3$^{rd}$-order accurate, meaning that the accuracy should improve as $\Delta x^3$ for well-resolved plumes. As we have seen above, this is far from the case for a long-lived plume in a divergent flow. The theoretical analysis of Rastigejev et al. (2010) indicates that the decay rate of a well-resolved plume should initially improve with the order of accuracy of the advection scheme, but that the rate of improvement should fall off to $\Delta x^{0.25-0.5}$ as plume stretching limits the ability to resolve gradients. Furthermore the decay rate is expected to eventually become grid-independent as the dimension of the filamented plume becomes comparable to the model grid (section 2). Here we use our ensemble of 2-D simulations ranging over a factor of 16 in grid resolution to evaluate that result, taking the plume decay rate constant *a* as a measure of accuracy.

Figure 5 shows the average improvement (reduction) in *a* with increasing grid resolution, for the plumes in different latitude bands and as a function of plume age. In the tropics, there is in general little flow divergence and numerical diffusion is weak as a result. Figure 5 shows that *a* in the fresh plume improves as $\Delta x^3$, consistent with the third-order accuracy of the PPM



advection scheme, and even in the aged plume the improvement scales as $\Delta x^2$. Thus we find that flow divergence does not limit the gains from increasing grid resolution, at least for these 2-D plumes. As shown in Fig. 3, a 1°×1.25° grid resolution seems sufficient to simulate long-range transport of tropical plumes with little numerical diffusion.

Outside the tropics, where flow divergence is greater, the effective order of accuracy is smaller and shows greater variability between grid resolutions with plume age. It starts second-order ($\Delta x^2$) but decreases as the plume ages and filaments. By day 5-6, the average order of convergence has decreased to $\Delta x^{0.5}$ for grid resolutions coarser than 1°×1.25°. At higher resolutions the rate of improvement with resolution is greater, albeit still of the order of $\Delta x$. There is curvature in the response of the plume decay constant to grid resolution, such that the benefit of increasing grid resolution increases as the resolution gets finer. This

is again in agreement with the theory of Rastigejev et al. (2010), where the purpose of increasing grid resolution is to drop below the scale where plume gradients are well resolved.

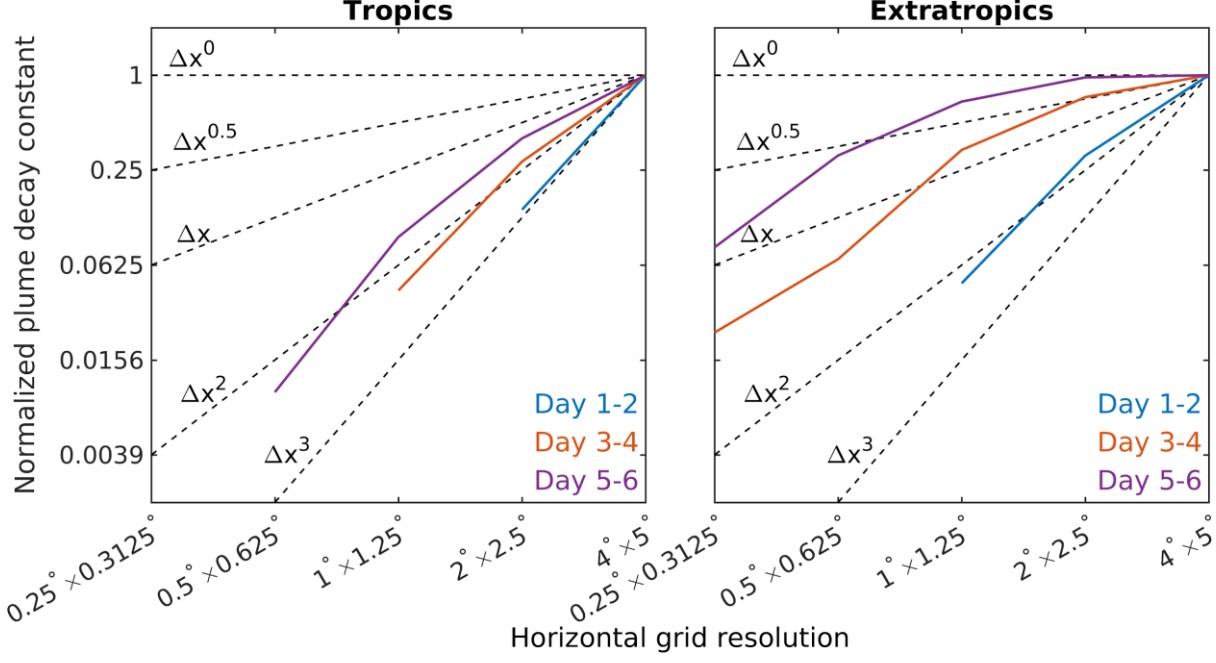

**Figure 5. Sensitivity of plume decay to grid resolution and its dependence on plume age. At each resolution, the average value of the plume decay constant *a* for each 48 hour period is calculated and divided by the value at the coarsest resolution (4°×5°), resulting in**
**the normalized decay constant shown as the ordinate. Results are shown for different plume aging times. The left-hand panel shows the average for the 36 plumes in the tropics, and the right-hand panel shows the average for the 54 plumes in the extratropics. The dashed lines show the order of improvement in the plume decay rate as a function of the grid resolution Δx, i.e. the effective order of accuracy of the advection scheme. Decay for tropical plumes at fine grid resolution is insignificant so that the plume decay constant is ill-defined, and not shown on this figure. This is also true of decay at fine resolutions in the first 2 days for extratropical plumes.**




## 6 3-D plume decay

We now turn to 3-D simulations for a more practical evaluation of the gains that could be made from increasing model resolution. An important distinction here is that we cannot explore a wide range of grid resolutions in the vertical; the ~0.6 km native vertical resolution of the GEOS-5 data in the free troposphere is comparable to the observed thickness of free

tropospheric plumes (Thouret et al. 2000). Such coarse resolution in the free troposphere seems typical of the current generation of models, which have emphasized improving horizontal resolution more than vertical resolution. For example, the ERA-Interim re-analysis, produced by the European Center for Medium-range Weather Forecasts (ECMWF), has a similar mean vertical resolution of 570 m in the free troposphere (Dee et al. 2011).

Global mean plume decay rates and plume sizes in 3-D are shown in Fig. 6 as a function of plume aging times, and compared to the 2-D cases discussed previously. In the 2-D simulations, each doubling of resolution yields a 10-20% improvement in the final maximum VMR after 9 days, up to a value of 89% at a resolution of 0.25°×0.3125°. The plume size improvement with increasing resolution is smaller but equally consistent. After 9 days, the plume size is double its initial size at 0.25°×0.3125°. In 3-D the numerical diffusion is considerably larger. At the 0.25°×0.3125° grid resolution, the maximum

VMR after 9 days drops to 13% of its original value, and the plume size increases by a factor of 6 from its original value. The reason is that the ability to preserve the plume is limited by numerical diffusion in the vertical, where the plume is initially poorly resolved in all cases because of the low native vertical resolution in the GEOS-5 fields.

There is also a counterproductive aspect to increasing horizontal resolution in a 3-D simulation. As the horizontal resolution

is increased, fine-scale vertical eddies are resolved that increase the vertical stretching of the plume, compromising the advantages gained from the slower horizontal diffusion. This is highlighted by the negligible improvement in plume size after 9 days between 3-D simulations at grid resolutions of 0.5°×0.625° and 0.25°×0.3125°. The increased vertical diffusion almost completely offsets the improvement provided by reducing spurious horizontal diffusion.





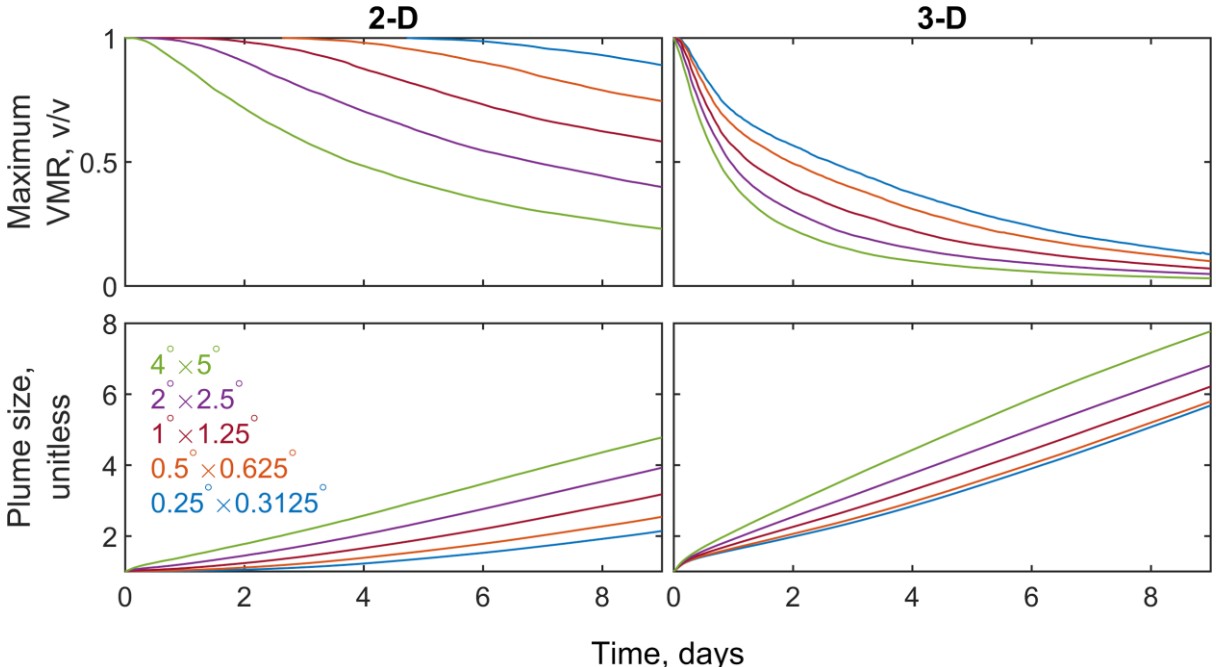

**Figure 6. Maximum VMR (top panel) and plume size (bottom panel) as a function of plume aging time. Values are global averages for the ensemble of 90 plumes in Fig. 1, and are shown for 2-D and 3-D simulations at the different horizontal grid resolutions indicated in the legend.**

Figure 7 summarizes the differences between the 2-D and 3-D results in the rate of plume decay as a function of grid resolution, latitude, and the flow divergence as measured by the Lyapunov exponent. The rate of improvement of the solution as the grid resolution increases is indicated in the figure by the vertical separation of points along each line relative to the maximum value. In 2-D, increasing resolution yields consistent benefits, such that simulations at or finer than 1°×1.25° achieve near-zero diffusion in the tropics. In 3-D, high rates of flow stretching remain correlated with higher rates of diffusion, but increasing the horizontal resolution yields a smaller relative decrease in the rate of diffusion, and the overall rate of convergence is of the order of $\Delta x^{0.25}$ or worse. We find that extratropical regions still consistently experience greater rates of numerical diffusion. However, even in the tropics where flow stretching is slow and 2-D simulations performed well, vertical diffusion due to poor vertical resolution provides a lower bound of $3\times10^{-6}$ s$^{-1}$ for $a$, corresponding to a decay time scale of 3 days. This shows that the ability of current global Eulerian models to resolve free tropospheric plumes is limited by the vertical grid resolution. Assuming isotropy in requirements between the horizontal resolution (which we were able to investigate in detail) and the vertical resolution, we conclude that a 100-m vertical resolution would be required to preserve the transport of free tropospheric plumes.





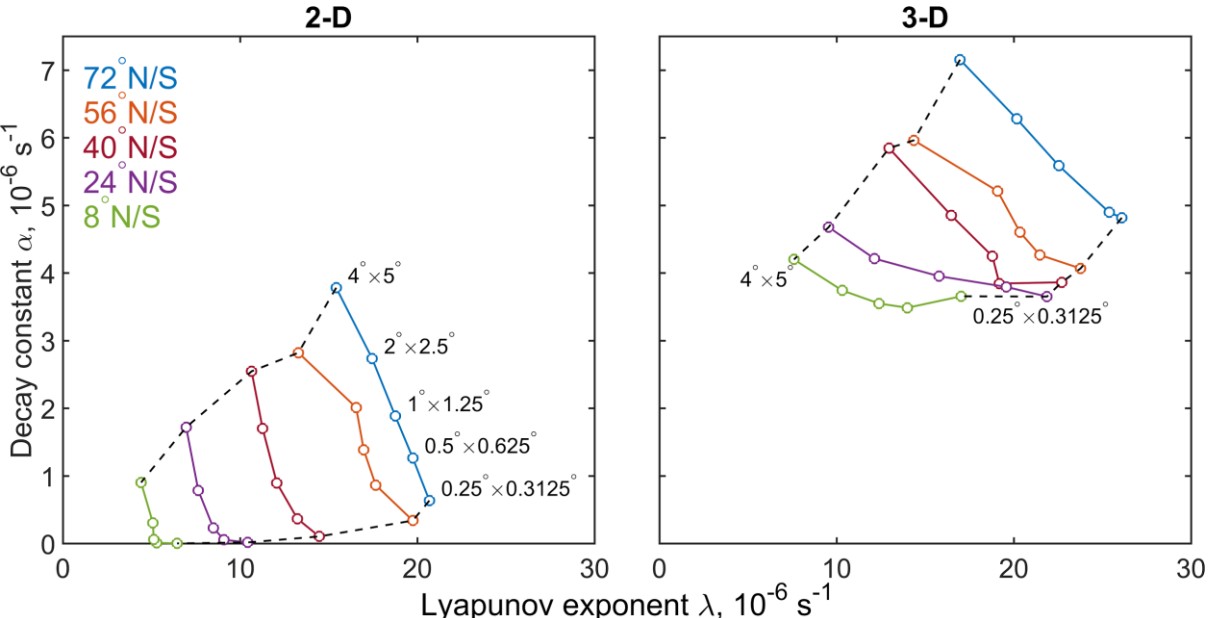

**Figure 7. Variation of the plume decay constant ($a$) in 2-D and 3-D simulations as a function of latitude and the Lyapunov exponent of the flow ($\lambda$). Each point shows the mean value of $a$ and $\lambda$ over the 9-day plume aging time in the free troposphere, averaging over all plumes at a given latitude in both hemispheres. Each colored line shows results from all 5 resolutions as individual circles. Dashed black lines show the effect of increasing absolute latitude for the resolution endpoints (0.25°×0.3125° and 4°×5°).**

## 7 Conclusions

We examined why global models are unable to simulate the intercontinental-scale transport of quasi-horizontal plumes in the free troposphere, dissipating them in a few days instead of preserving their coherence. Our focus was to test theoretical results by Rastigejev et al. (2010) that this dissipation is due to fast numerical diffusion in the divergent flow typical of the free troposphere. Divergence (shear, stretching) causes the plume to filament rapidly to the point when it is not properly resolved on the model grid. At that point fast dissipation takes place in the model regardless of the accuracy of the advection scheme and only weakly dependent on grid resolution.

We conducted a large worldwide ensemble of simulations of free tropospheric plumes with the GEOS-Chem chemical transport model driven by NASA GEOS-5 assimilated meteorological data. The simulations used horizontal resolutions ranging from 0.25°×0.3125° (native GEOS-5) to 4°×5°, including only advection (no subgrid turbulence or chemistry). Restriction to advection allowed us to focus on numerical diffusion – in a purely advective problem the plume should not dilute, even in a divergent flow. We diagnosed plume decay caused by numerical diffusion in the model by the decrease in maximum volume mixing ratio (VMR) in the plume (decay rate constant $a$) and by the increase in plume size. Native vertical resolution in GEOS-5 (and other current meteorological models) is ~0.5 km in the free troposphere, too coarse to adequately resolve vertical gradients in plumes (typically ~1 km thick). Thus we conducted both 2-D (horizontal) and 3-D simulations.



Restriction to 2-D allowed us to investigate in detail the sensitivity to grid resolution for initially well-resolved plumes (12°×15°). Extension to 3-D allowed us to examine numerical diffusion in realistic model situations.

We find that extratropical plumes decay much faster than tropical plumes, and that this can be explained by stronger flow divergence measured by the Lyapunov exponent of the flow ($\lambda$). Under strongly divergent flow typical of the extratropics, the rate of plume decay varies linearly with $\lambda$ and improves only as the square root of the grid resolution $\Delta x$. We find that the sensitivity of the plume decay to grid resolution decreases as the plume ages, initially improving as $\Delta x^3$ (the order of accuracy of the GEOS-Chem advection scheme), and eventually decaying after a few days to $\Delta x^2$ in the tropics and $\Delta x^{0-1}$ in the extratropics.

3-D plume decay in our simulations is much faster than in 2-D, and consistent with the general inability of models to preserve the coherence of free tropospheric plumes. The plume decay rate in 3-D still depends on horizontal flow divergence, but the sensitivity to horizontal grid resolution is weaker and the decay is instead limited by the coarse vertical resolution. Vertical numerical diffusion is very fast, and is amplified at finer horizontal resolution by vertical eddies that would be smoothed out at coarser horizontal resolution. Even tropical plumes decay with a time constant of about 3 days.

Thus we find that increasing vertical grid resolution in the free troposphere to ~100 m is an essential first step for models to resolve the intercontinental-scale transport of free tropospheric plumes. Increasing horizontal resolution beyond 1° is futile. Even then, the modeling problem will remain challenging in strongly divergent flow typical of high latitudes. More advanced solutions might involve adaptive grids designed to resolve local conditions of large chemical gradients and large divergence, or embedding Lagrangian plumes in the global Eulerian modeling framework.

**Author contributions**

SDE and DJJ designed the experiments. SDE developed model code and performed all experiments. SDE prepared the manuscript in collaboration with DJJ.

**Competing interests**

The authors declare that they have no conflict of interest.





**Acknowledgements**

The authors would like to thank Christopher Holmes for discussion and for advice on producing divergence-free, 2-D atmospheric flows. We would also like to thank Meiyun Lin for discussion on the significance of numerical diffusion in modern GCMs. The GEOS-5 FP meteorological data used in this study have been provided by the Global Modeling and Assimilation Office (GMAO) at NASA Goddard Space Flight Center. This research was supported by the NOAA Climate and Global Change Postdoctoral Fellowship Program, administered by UCAR's Visiting Scientist Programs. SDE was also supported by a Harvard University Center for the Environment (HUCE) Fellowship. DJJ was supported by the NASA Earth Science Division.

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
