# Peer review of "Limits on the ability of global Eulerian models to resolve intercontinental transport of chemical plumes"

_Atmospheric Chemistry and Physics, 2016_

## Referee Comment (RC1) · A. Stohl (Referee) · 1 Dec 2016

This is an interesting study that I can fully recommend for publication. The paper addresses an important aspect of chemical transport modelling that probably most modelers are somewhat aware of (or maybe not), but often seem to prefer ignoring because of its inconvenience. I already liked the paper of Rastigejev et al. (2010) and this paper from the same group dwells deeper into the problems found by these authors. The paper is clear, concise and well written. I congratulate the authors for being critical about an issue that may not be very popular but is nevertheless important. The following comments may be considered by the authors when revising their paper.

Page2, lines 11-17: This text is a bit unbalanced. It is true that Lagrangian models have

difficulties with non-linear chemistry, but that is true both in the troposphere and stratosphere, and not only in the troposphere. On the other hand, convective motions are not a problem at all (Forster et al., 2007). Convective schemes have been implemented in several Lagrangian models, so transport in the troposphere can be well represented. In fact, there are hundreds, if not thousands of papers in the literature using such models for studying long-range transport in the troposphere, even such using models without a convection scheme.

Page 6, line 3: The model uses meteorological data from an assimilation system. This can be problematic in itself for tracer advection. As shown by Stohl et al. (2004) using a trajectory model, dynamical inconsistencies due to the data assimilation lead to increased diffusivity. It would be interesting to know how important this is for the results obtained by the authors. A possibility for this would be to run GEOS-Chem on a dynamically consistent meteorological data set from a free-running simulation, not using data assimilation. Some of the diffusion the authors find may not really be due to the Eulerian advection scheme, but to the dynamically inconsistent input data used.

Page 9, lines 1-4: It is argued that plumes in the tropics are better preserved. This is true as a function of time. On the other hand, transport speeds of plumes in the tropics are typically slower and that means that, relative to distance travelled, the plumes probably diffuse at a similar rate. For intercontinental transport that probably means that plumes in the tropics are not more coherent than in the extratropics once they reach a downwind continent at a similar distance as in the extratropics. Both transport and diffusion just take a longer time.

Page 18, lines 17-18: The statement "Thus we find that increasing vertical grid resolution in the free troposphere to ∼100 m is an essential first step for models to resolve the intercontinental-scale transport of free tropospheric plumes." is not really supported by the analysis. While this is a good suggestion, it is not really a finding of this study but rather an extrapolation that would need testing to be called a finding. Thus, I suggest a more careful phrasing.

**[ACPD]{.underline}**

Interactive
comment

Minor points, language: Page 1, line 23: "affect intercontinental scales". This is a strange wording, as the scales are not affected but plume coherence over those scale is affected.

Page 8, line 3: "artifact information": I would recommend removing "information" as it is not clear what the word information refers to.

Page 11, line 9: "after 9 days 1x1.25" should be "after 9 days at 1x1.25 resolution."

Page 12, line 16: "plumes being to decay" should be "plumes begin to decay".

References:

Forster, C., A. Stohl, and P. Seibert (2007): Parameterization of convective transport in a Lagrangian particle dispersion model and its evaluation. J. Appl. Met. Clim. 46, 403-422.

Rastigejev, Y., Park, R., Brenner, M. P. and Jacob, D. J.: Resolving intercontinental pollution plumes in global models of atmospheric transport, J. Geophys. Res., 115, D02302, 2010.

Stohl, A., O. Cooper, and P. James (2004): A cautionary note on the use of meteorological analysis data for quantifying atmospheric mixing. J. Atmos. Sci. 61, 1446-1453.

---

## Short Comment (SC1) · 14 Dec 2016

This is an interesting paper on the influence of model resolution on intercontinental transport of chemical plumes. The paper is overall well written. However, I believe there is an inadequate recognition of previous work that are based on models other than GEOS-Chem. This is probably an oversight, but almost all studies cited in the Introduction of the current manuscript are based on one single model - GEOS-Chem. On Page 2, about Line 5: the authors stated that "Eulerian models used for simulating global atmospheric transport fail to reproduce this persistent layered structure". However, there are a few studies showing that high-resolution models with interactive stratospheric and tropospheric chemistry have skills simulating the layered structure of

ozone plumes in the free troposphere.

For example, Lin et al. (2012A, JGR) showed that the GFDL-AM3 model at ∼50km horizontal resolution successfully reproduces observed sharp ozone gradients above California, including the interleaving and mixing of Asian pollution and stratospheric air associated with complex interactions of midlatitude cyclone air streams (see their Figures 2 and 5 for comparison with ozonesondes).

A follow up paper by Lin et al. (2012B, JGR) showed that GFDL-AM3 captures the observed layered features and sharp ozone gradients of deep stratospheric intrusions over the western United States (see their Figures 3, 5, and 7).

Lin et al. (2015, Nature Communications) further examined the influence of horizontal resolution in GFDL-AM3 on the simulation of deep stratospheric intrusions (see their Methods and Supplementary Figures 1 to 2).

These models are obviously not perfect. For example, Lin et al. (2012B) found that the layers with the peak O3 enhancements in the model appear to be wider in thickness and lower in altitude than observed by the sondes. This discrepancy will actually support your idea to increase the model vertical resolution as well.

References:

Lin2012A: Lin, Meiyun, A. M. Fiore, L. W. Horowitz, O. R. Cooper, V. Naik, J. Holloway, B. J. Johnson, A. M. Middlebrook, S. J. Oltmans, I. B. Pollack, T. B. Ryerson, J. X. Warner, C. Wiedinmyer, J. Wilson, B. Wyman: Transport of Asian ozone pollution into surface air over the western United States in spring , Journal of Geophysical Research,117, D00V07, 2012, doi:10.1029/2011JD016961

Lin2012B: Lin Meiyun, A. M. Fiore , O. R. Cooper , L. W. Horowitz , A. O. Langford , Hiram Levy II , B. J. Johnson , V. Naik , S. J. Oltmans , C. Senff (2012): Springtime high surface ozone events over the western United States: Quantifying the role of stratospheric intrusions, Journal of Geophysical Research, 117, D00V22,

doi:10.1029/2012JD018151

Meiyun Lin, A.M. Fiore, L.W. Horowitz, A.O. Langford, S. J. Oltmans, D. Tara­sick, H.E. Reider (2015): Climate variability modulates western US ozone air qual­ity in spring via deep stratospheric intrusions, Nature Communications, 6, 7105, doi:10.1038/ncomms8105

---

## Referee Comment (RC2) · Anonymous Referee #2 · 19 Dec 2016

General comments: This paper discusses the ability of the GEOS-Chem Eulerian model to resolve long range transport of chemical plumes in the free troposphere by simulating atmospheric transport in idealized 2D and 3D cases in which only advection is considered. Different metrics such as maximum volume mixing ratio, plume size are used to estimate a plume decay constant. Free tropospheric plumes decay much faster in the mid-latitudes than in the tropics because of stronger divergent flow. Sensitivity to the horizontal model resolution is discussed. The 3D simulations shows that the limiting factor in Eulerian chemical transport models capability to resolve free tropospheric plumes is more the vertical resolution than the horizontal resolution. The authors suggest a vertical resolution of 100m to preserve free tropospheric plumes in

Eulerian models.

The paper is well written and the results are of interest for the community. I recommend this paper for publication after addressing the following minor comments.

Main comments: Introduction: page 2, line 13-14: Lagrangian models have been used in numerous publications to describe long range transport of plumes originating from the boundary layer, free troposphere and stratosphere, with or without convection. This sentence should be rephrased or removed.

3D plume decay: page 15: The authors suggest that increasing the horizontal resolution in the model increases the development of fine scale vertical eddies and hence increases diffusion. Is there a meteorological product in GEOS that supports this claim?

Vertical resolution: page 16: Line 15 to 18: The conclusion on the vertical resolution is an important result, but the explanations given here are not convincing. I would like to see more details.
* * *

---

## Author Comment (AC1) · 29 Jan 2017

**Sebastian D. Eastham**
**Postdoctoral Researcher**
**Joint NOAA and Harvard Uni. Center for the Env. Fellow**
**Harvard Atmospheric Chemistry Modeling Group**

[Figure]

**Harvard University**
**29 Oxford Street, Pierce Hall 110D**
**Cambridge, Massachusetts 02138, USA**
**http://scholar.harvard.edu/seastham**

Atmospheric Chemistry and Physics Editorial Office

29 January 2017

Dear Editor,

**Re: Submission of "Limits on the ability of global Eulerian models to resolve intercontinental transport of chemical plumes" to *Atmospheric Chemistry and Physics***

Thank you for considering our submission and for arranging the careful reviews. Our revised manuscript is enclosed; please find below our responses to each of the comments made by the reviewers. We have also included a response to the short comment from Dr. Meiyun Lin. Finally, several minor adjustments have been made for the sake of clarity, mostly to the introduction. These changes have no effect on the findings of the paper and are shown in the highlighted manuscript.

*Reviewer #1 (Dr. Andreas Stohl)*

This is an interesting study that I can fully recommend for publication. The paper addresses an important aspect of chemical transport modelling that probably most modelers are somewhat aware of (or maybe not), but often seem to prefer ignoring because of its inconvenience. I already liked the paper of Rastigejev et al. (2010) and this paper from the same group dwells deeper into the problems found by these authors. The paper is clear, concise and well written. I congratulate the authors for being critical about an issue that may not be very popular but is nevertheless important. The following comments may be considered by the authors when revising their paper.

Page2, lines 11-17: This text is a bit unbalanced. It is true that Lagrangian models have difficulties with non-linear chemistry, but that is true both in the troposphere and stratosphere, and not only in the troposphere. On the other hand, convective motions are not a problem at all (Forster et al., 2007). Convective schemes have been implemented in several Lagrangian models, so transport in the troposphere can be well represented. In fact, there are hundreds, if not thousands of papers in the literature using such models for studying long-range transport in the troposphere, even such using models without a convection scheme.

**We have revised this sentence to more fairly demonstrate why purely Lagrangian models are not appropriate to the problem at hand. We no longer refer to convection and have changed the phrasing to highlight the issues of homogeneous global coverage and non-linear chemistry, for which Eulerian models are the usual solution.**

Page 6, line 3: The model uses meteorological data from an assimilation system. This can be problematic in itself for tracer advection. As shown by Stohl et al. (2004) using a trajectory model, dynamical inconsistencies due to the data assimilation lead to increased diffusivity. It would be interesting to know how important this is for the results obtained by the authors. A possibility for this would be to run GEOS-Chem on a dynamically consistent meteorological data set from a free-running simulation, not using data assimilation. Some of the diffusion the authors find may not really be due to the Eulerian advection scheme, but to the dynamically inconsistent input data used.

**Stohl et al. (2004) demonstrates an interesting additional component to the problem. However, a full and fair assessment of the impact of physical inconsistencies due to data assimilation would be a significant undertaking and deserves a more thorough assessment than we feel we can give in the paper at this point. We have added a statement in the introduction which clarifies that the use of data assimilation is a factor in producing the observed numerical diffusion.**

Page 9, lines 1-4: It is argued that plumes in the tropics are better preserved. This is true as a function of time. On the other hand, transport speeds of plumes in the tropics are typically slower and that means that, relative to distance travelled, the plumes probably diffuse at a similar rate. For intercontinental transport that probably means that plumes in the tropics are not more coherent than in the extratropics once they reach a downwind continent at a similar distance as in the extratropics. Both transport and diffusion just take a longer time.

**This is consistent with our findings, and we agree that distance traveled is a useful independent variable to consider. Furthermore, the lower wind speeds in the tropics will also correlate with lower wind shear, and therefore are important to understanding the cause of the reduced Lyapunov exponents observed there. However, we found that the latitudinal gradient in rate of diffusion is similar when plotting the data as a function of distance traveled. Figure C1 (below) shows the peak mixing ratio as a function of time (top) and as a function of distance travelled (bottom) for plumes at different latitudes in the 2-D simulations at 1°×1.25° grid resolution. While it is true that tropical plumes travel a shorter distance due to the lower wind speeds and that this complicates the analysis, Figure C1 shows that a plume which has travelled (for example) 6,000 km after beginning at 70° S will have decayed more than a plume which originated at 40° S. We have added a sentence to the paragraph in question to reflect this.**

[Figure]

*Figure C1: Maximum mixing ratio in a 2-D simulation. The upper plot shows the average for each latitude band as a function of travel time. The lower plot shows the average for each latitude band as a function of the approximate distance traveled by the plume. Due to the lower wind speeds in tropical regions, the final distance traveled within 9 days is smaller (~3,500 km).*

Page 18, lines 17-18: The statement "Thus we find that increasing vertical grid resolution in the free troposphere to ~100 m is an essential first step for models to resolve the intercontinental-scale transport of free tropospheric plumes." is not really supported by the analysis. While this is a good suggestion, it is not really a finding of this study but rather an extrapolation that would need testing to be called a finding. Thus, I suggest a more careful phrasing.

**We have softened the wording in this paragraph to reflect that our recommended vertical resolution is an extrapolation rather than a fully validated result. We have also extended our explanation of the recommendation at the end of section 6 (see response to reviewer 2).**

*Minor points, language*: Page 1, line 23: "affect intercontinental scales". This is a strange wording, as the scales are not affected but plume coherence over those scale is affected.

**This line has been changed. Previously, the sentence read: "Much of this transport takes place in well-defined, concentrated layers or plumes that can remain coherent for a week or more and affect intercontinental scales". This has been modified to read: "Much of this transport takes place in well-defined, concentrated layers or plumes that can remain coherent for a week or more *while traveling over distances of intercontinental scale*".**

Page 8, line 3: "artifact information": I would recommend removing "information" as it is not clear what the word information refers to.

**We agree, and the line has been changed accordingly.**

Page 11, line 9: "after 9 days 1x1.25" should be "after 9 days at 1x1.25 resolution."

**This has been changed accordingly.**

Page 12, line 16: "plumes being to decay" should be "plumes begin to decay".

**This has been changed accordingly.**

References:  Forster, C., A. Stohl, and P. Seibert (2007): Parameterization of convective transport in a Lagrangian particle dispersion model and its evaluation. J. Appl. Met. Clim. 46, 403-422.

Rastigejev, Y., Park, R., Brenner, M. P. and Jacob, D. J.: Resolving intercontinental pollution plumes in global models of atmospheric transport, J. Geophys. Res., 115, D02302, 2010.

Stohl, A., O. Cooper, and P. James (2004): A cautionary note on the use of meteorological analysis data for quantifying atmospheric mixing. J. Atmos. Sci. 61, 1446-1453.

*Reviewer #2 (Anonymous)*

This paper discusses the ability of the GEOS-Chem Eulerian model to resolve long range transport of chemical plumes in the free troposphere by simulating atmospheric transport in idealized 2D and 3D cases in which only advection is considered. Different metrics such as maximum volume mixing ratio, plume size are used to estimate a plume decay constant. Free tropospheric plumes decay much faster in the mid-latitudes than in the tropics because of stronger divergent flow. Sensitivity to the horizontal model resolution is discussed. The 3D simulations shows that the limiting factor in Eulerian chemical transport models capability to resolve free tropospheric plumes is more the vertical resolution than the horizontal resolution. The authors suggest a vertical resolution of 100m to preserve free tropospheric plumes in Eulerian models. The paper is well written and the results are of interest for the community. I recommend this paper for publication after addressing the following minor comments.

*Main comments*: Introduction: page 2, line 13-14: Lagrangian models have been used in numerous publications to describe long range transport of plumes originating from the boundary layer, free troposphere and stratosphere, with or without convection. This sentence should be rephrased or removed.

**We have revised this phrasing (see response to reviewer 1) to more fairly represent the Lagrangian approach and to better explain why we focus on Eulerian models.**

3D plume decay: page 15: The authors suggest that increasing the horizontal resolution in the model increases the development of fine scale vertical eddies and hence increases diffusion. Is there a meteorological product in GEOS that supports this claim?

**This is inevitably true due to the method of degrading horizontal resolution used in this study. This effect has also been observed in previous work, such as Wild and Prather (2006) which was itself cited by Rastigejev et al (2010).**

Vertical resolution: page 16: Line 15 to 18: The conclusion on the vertical resolution is an important result, but the explanations given here are not convincing. I would like to see more details.

**The 100 m recommendation stems from the fact that a factor of 4 increase in horizontal resolution yielded effectively zero diffusion in tropical plumes. We therefore apply the same factor to vertical resolution to provide a recommendation. We now state this explicitly, and have added a caveat that this recommendation is made pending a dedicated investigation.**

**Reference: Wild, O. and Prather, M. J.: Global tropospheric ozone modeling: Quantifying errors due to grid resolution, Journal of Geophysical Research, 111(D11), doi:10.1029/2005JD006605, 2006.**

*Short comment from Dr. Meiyun Lin*

This is an interesting paper on the influence of model resolution on intercontinental transport of chemical plumes. The paper is overall well written. However, I believe there is an inadequate recognition of previous work that are based on models other than GEOS-Chem. This is probably an oversight, but almost all studies cited in the Introduction of the current manuscript are based on one single model - GEOS-Chem. On Page 2, about Line 5: the authors stated that "Eulerian models used for simulating global atmospheric transport fail to reproduce this persistent layered structure". However, there are a few studies showing that high-resolution models with

interactive stratospheric and tropospheric chemistry have skills simulating the layered structure of ozone plumes in the free troposphere.

For example, Lin et al. (2012A, JGR) showed that the GFDL-AM3 model at ~50km horizontal resolution successfully reproduces observed sharp ozone gradients above California, including the interleaving and mixing of Asian pollution and stratospheric air associated with complex interactions of midlatitude cyclone air streams (see their Figures 2 and 5 for comparison with ozonesondes).

A follow up paper by Lin et al. (2012B, JGR) showed that GFDL-AM3 captures the observed layered features and sharp ozone gradients of deep stratospheric intrusions over the western United States (see their Figures 3, 5, and 7).

Lin et al. (2015, Nature Communications) further examined the influence of horizontal resolution in GFDL-AM3 on the simulation of deep stratospheric intrusions (see their Methods and Supplementary Figures 1 to 2). These models are obviously not perfect. For example, Lin et al. (2012B) found that the layers with the peak O3 enhancements in the model appear to be wider in thick- ness and lower in altitude than observed by the sondes. This discrepancy will actually support your idea to increase the model vertical resolution as well.

Thank you again for considering this work for *Atmospheric Chemistry and Physics*. We would like to also thank the reviewers for their time and their comments. The changes we have made in response to their concerns are highlighted in an attached copy of the manuscript. Additions are highlighted in blue, and deletions in red.

Sincerely,

Sebastian D. Eastham

[revised manuscript text omitted]